# Efficacy of Tocilizumab in Management of COVID-19 Patients Admitted to Intensive Care Units: A Multicenter Retrospective Cohort Study

**DOI:** 10.3390/medicina59010053

**Published:** 2022-12-27

**Authors:** Hoda Younes Ibrahem, Doaa Hamdy Aly, Ahmed E. Abou Warda, Ramadan Abdelmoez Farahat, Raghda Mamdouh Youssef, Mona Hassan Abdelhamid, Heba Ahmed Goud, Rana Ragab Mohamed, Menna Allah Y. Nasr Eldien, Fahad Obaid Alotaibi, Abdulaziz Ibrahim Alzarea, Abdullah Salah Alanazi, Nehal M. Eisa, Abdelrahman SH. Refaee

**Affiliations:** 1Clinical Research Department at Giza Health Affairs Directorate, Ministry of Health and Population, Giza P.O. Box 12511, Egypt; 2Clinical Pharmacy Department, Faculty of Pharmacy, October 6 University, Giza P.O. Box 12585, Egypt; 3Faculty of Medicine, Kafrelsheikh University, Kafr Elsheikh P.O. Box 33511, Egypt; 4Forensic Medical Services Center in Riyadh, Ministry of Health, Riyadh 12271, Saudi Arabia; 5Clinical Pharmacy Department, College of Pharmacy, Jouf University, Sakaka 72388, Saudi Arabia; 6Health Sciences Research Unit, Jouf University, Sakaka 72388, Saudi Arabia

**Keywords:** COVID-19, Tocilizumab, cytokine release syndrome, in-hospital mortality

## Abstract

*Background and Objectives*: Mortality and illness due to COVID-19 have been linked to a condition known as cytokine release syndrome (CRS) that is characterized by excessive production of inflammatory cytokines, particularly interleukin-6 (IL-6). Tocilizumab (TCZ), a recent IL-6 antagonist, has been redeployed as adjunctive treatment for CRS remission in COVID-19 patients. This study aimed to determine the efficacy of Tocilizumab on patients’ survival and the length of stay in hospitalized COVID-19 patients admitted to the intensive care unit. *Methods*: Between January 2021 and June 2021, a multicenter retrospective cohort study was carried out in six tertiary care hospitals in Egypt’s governorate of Giza. Based on the use of TCZ during ICU stay, eligible patients were divided into two groups (control vs. TCZ). In-hospital mortality was the main outcome. *Results*: A total of 740 patient data records were included in the analysis, where 630 patients followed the routine COVID-19 protocol, while 110 patients received TCZ, need to different respiratory support after hospitalization, and inflammatory mediators such as C-reactive protein (CRP), ferritin, and Lactate dehydrogenase (LDH) showed a statistically significant difference between the TCZ group and the control group. Regarding the primary outcome (discharged alive or death) and neither the secondary outcome (length of hospital stay), there is no statistically significant difference between patients treated with TCZ and the control group. *Conclusions*: Our cohort of patients with moderate to severe COVID-19 did not assert a reduction in the risk of mortality or the length of stay (LOS) after TCZ administration.

## 1. Introduction

The coronavirus SARS-CoV-2 has been at the center of a worldwide pandemic that has plagued the world since December 2019 [1,2]. Patients infected with COVID-19 have a 75% chance of completing their recovery without major complications, but a 25% probability that they may become critically ill and require intensive care unit (ICU) treatment or even ending with death [3]. COVID-19 can cause a wide variety of symptoms and signs, from no symptoms at all to life-threatening pneumonia and respiratory failure causing acute respiratory distress syndrome (ARDS), leading to a high need for supplemental oxygen. It is possible for the disease to progress from its first phase of viral replication into a second phase driven by the host inflammatory response, resulting in the rapid onset of severe respiratory failure. Typical radiological data implies that infection with coronavirus 2 (SARS-CoV-2) may trigger a hyperimmune response associated with acute respiratory distress syndrome. Patients in the worst shape can experience something called a “cytokine release storm” [4].

Cytokine release syndrome (CRS) has been reported to cause COVID-19 related morbidity and mortality. CRS is due to the excess release of proinflammatory cytokines, especially Interleukin-6 (IL-6) [5]. IL-6 causes respiratory dysfunction including severe alveolar damage, alveolar-capillary leakage, and impaired blood–gas exchange (especially oxygen diffusion) [6]. Recent IL-6 antagonists, especially tocilizumab, have been repurposed for the remission of CRS in COVID-19 patients as adjunctive therapy where multiple case series have suggested a potential role for tocilizumab.

Tocilizumab (TCZ) is a monoclonal antibody against the IL-6 receptor that has found widespread application in the treatment of rheumatic illnesses and severe Chimeric Antigen Receptor (CAR) T cell-induced CRS around the world. Treatment of critically ill patients with COVID-19 may benefit from TCZ because of its potent influence on the inflammatory response brought on by the virus. However, it is important to use TCZ at the right points in the treatment plan [7].

There was large variation in the results of the previous studies that range from effect on mortality, time to clinical improvement, and prevention of progression to a more severe form of the disease [8]. This indicates that the standardization of the tocilizumab dose or the stage of the disease at which the drug is administered have not been well established, as well as the type of patients that will benefit from this drug. Other factors that may contribute in the pathophysiology of the disease are the patient’s characteristics and the presence of different SARS-CoV-2 strains [9]. By providing an estimate of the immunomodulatory agent’s possible effects, and for the high cost of tocilizumab, which in turn may be a burden on the healthcare system, observational studies may aid in the design of randomized clinical trials for the remission of critically-ill COVID-19 patients [10]. Thus, we aim to assess the efficacy of tocilizumab in the remission of COVID-19 patients hospitalized in the ICU. We also aim to assess the effect of tocilizumab on the length of the hospital stays and the result of co-administration of tocilizumab with other treatments.

## 2. Methods

### 2.1. Study Design and Seting

From January 2021 to June 2021, a multi-center retrospective cohort study was conducted on the intensive care units of six hospitals in Giza Governorate, Egypt. Among these, Al Tahrir, Om Elmasryeen, and Al Hawamdyia are the three General Hospitals, and the Central Hospitals include Al Warraq, Six of October, and Elsheikh Zayed. The Declaration of Helsinki and World Health Organization recommendations were all adhered to throughout the current study. The research protocol has been reviewed and approved by the “Research Ethics Committee” of the Central Directorate for Research and Health Development in MOHP (REC No. 23-2021/14).

### 2.2. Subjects

The study included all patients with a diagnosis of moderate to severe COVID-19 pneumonia who were admitted to the intensive care units of Giza hospitals. The study excluded patients with mild COVID-19, patients under the age of 18, patients who died within 48 h of being admitted, pregnant women, patients who were breastfeeding, and patients with mild COVID-19 symptoms.

### 2.3. Sampling Technique and Method of Selection

Consecutive sampling techniques were applied. Certain hospitals in Giza Governorate were selected to collect data from the patient records. All patients of each hospital who fulfilled the inclusion criteria were chosen. Data of 112 tocilizumab administrated patients were collected, and 2 of them were excluded for not fulfilling the criteria. On the other hand, 1257 data records were obtained from patients who were taking the routine protocol. Only 630 were included in the study, which was due to duplication of entry, entry errors, not fulfilling inclusion criteria, and incomplete data records.

### 2.4. Data Collection Tools

We used a predesigned structured questionnaire to collect the following data from the hospitals. The questionnaire was organized as follows: (Form A) covered patient identifiers, age, sex, coexisting conditions, investigations performed at the time of admission, treatments received in the hospitals and patients’ outcomes. (Form B) covered TCZ-related conditions: Criteria of severity of COVID-19 associated cytokine storm syndrome, contraindication, and side effects of TCZ.

### 2.5. Treatment Protocol and Outcomes

COVID-19 patients were tested using a COVID-19 laboratory panel consisting of C-reactive protein (CRP), ferritin, d-dimer, lactate dehydrogenase, and troponin I, as per institutional procedure. Tocilizumab treatment was given to patients who met the following requirements: symptoms of respiratory compromise including tachypnea, dyspnea, OR peripheral capillary oxygen saturation (SpO2) 90% on at least 4 L of oxygen OR rising oxygen requirements over 24 h, PLUS 2 or more of the following predictors for severe disease: C-reactive protein level higher than 35 mg/L, a ferritin level over 500 ng/mL, a D-dimer level over 1 mcg/L, a neutrophil-lymphocyte ratio over 4, or a lactate dehydrogenase level over 200 U/L.

All patients received a single IV infusion of 400 mg of tocilizumab as part of the therapy protocol. After tocilizumab delivery, antimicrobial prophylaxis was not consistently administered [11]. The patients’ clinical status was evaluated as the main outcomes, including being alive at discharge and death rate, and the secondary outcome including time to hospital discharge and predictors of mortality were also analyzed in the study.

### 2.6. Statistical Analysis

Data management and statistical analysis were performed using the Statistical Package for Social Sciences (SPSS) version 24(SPSS Inc, Chicago, IL). After being checked for accuracy, the obtained data were recoded, entered into a database, and statistically evaluated using the proper statistical tools and tests. Means, standard deviations, or medians and ranges were used to summarize numerical data. The Shapiro–Wilk test and Kolmogorov–Smirnov test were used to determine the data normality. Percentages were used to summarize categorical data. In quantitative variables, comparisons between patient groups were made using the independent t-test. Fisher’s exact tests and χ2 (chi square) tests were used to assess differences for categorical variables. A multivariate logistic regression was used to evaluate factors affecting primary outcomes. Kaplan–Meier curve and log rank test were used for survival analysis. *p*-values are always two-sided. *p*-values below 0.05 were deemed significant, and the Bonferroni technique was used to account for multiplicity.

## 3. Results

### 3.1. Baseline Characteristics of the Study Population

Out of the 740 patients recruited in the study, 630 patients followed the routine COVID-19 protocol while 110 patients received TCZ. The TCZ group had a median age of 58.5, whereas the control group’s median age was 63. Male patients accounted for a slightly larger proportion of the patient number in both groups, at 68 (61.8%) and 283 (44.9%). Regarding the severity of the disease, the proportions of patients with moderate severity were 69 (62.7%), 441 (70%) and severe illness were 41 (37.3%), 189 (30%) in the TCZ and control, respectively, without any statistically significant difference.

For all comorbidities, there was no significant difference between the TCZ and the control group except for cardiovascular diseases (*p* = 0.009), as shown in Table 1. Indeed, D dimer, creatinine, complete blood count, lymphopenia, lymphocytosis, thrombocytopenia, thrombocytosis, leukopenia, and leukocytosis all showed no statistically significant differences between the TCZ group and the control group, as illustrated in Table 2. In addition, all the recruited patients received standard treatments during hospitalization according to the Egyptian protocol for COVID-19 management, The different treatment groups were illustrated in Table 3. Additionally, at hospital admission, the two groups’ CT scan results, including ground glass opacity, infiltrations, and patchy shadowing, were compared, as indicated in Table 4.

### 3.2. Effect on the Clinical Outcomes

The primary outcome (death rate) shows a higher rate in the TCZ group without any statistically significant difference with the control group (*p* = 0.297), while the secondary outcome (length of hospital stay) shows a significant difference in favor of the control group (*p* = 0.045) as shown in Table 5. Furthermore, a comparison of LOS between the two groups using a Kaplan–Meier curve and a log rank test shows no statistically significant difference (*p* = 0.6690), as shown in Figure 1. Regarding the correlation between patients who required respiratory support and who were either discharged alive or died, it was shown that invasive and noninvasive ventilation had a greater death rate than other respiratory support systems, as indicated in Table 6.

A multivariant logistic regression analysis revealed that the only factors associated with mortality were COVID-19 severity (OR = 1.536, C.I. = 1.022–2.309, *p* = 0.039), CT scan findings (OR = 1.562, C.I. = 0.434–5.627, *p* = 0.045), Thrombocytosis (OR = 2.833, C.I. = 1.053–7.627, *p* = 0.039), Ivermectin (OR = 1.605, C.I. = 1.093–2.356, *p* = 0.016) Convalescent plasma (OR = 14.838, C.I. = 1.256–17.298, *p* = 0.032), Antifungals (OR = 0.242, C.I. = 0.090–0.650, *p* = 0.005), and Respiratory support (OR = 0.422, C.I. = 0.360–0.495, *p* = 0.001), as shown in Table 7.

## 4. Discussion

Cytokine storm was thought to be a significant factor in COVID-19 exacerbation and even death, since it can cause immunological dysregulation, lung tissue damage, hypoxia, and even respiratory failure [12,13]. Tocilizumab was reported to regulate the cytokine storm and inflammation in COVID-19 [14,15].

Even though our study did not find any advantages of tocilizumab over the conventional treatment in terms of mortality or length of hospital stay, other observational studies have shown better outcomes. The two largest observational studies to date on this topic have demonstrated an association between tocilizumab use and decreased mortality [16,17]. Guaraldi et al. evaluated the impact of tocilizumab regardless of the time of administration on a cohort of 1351 COVID-19 patients [16]. However, invasive ventilation or death was the composite endpoint, and patients admitted to the intensive care unit were not included. In the STOP-COVID tocilizumab study, which comprised 3924 severely ill COVID-19 patients admitted to ICUs, patients who received tocilizumab within the first two days of admission had a lower mortality risk than those who did not get tocilizumab within the first two days of admission [17]. In contrast to these two trials, the largest cohort study of 544 COVID-19 patients, carried out by Cardona-Pascual et al., supports our findings that tocilizumab treatment did not lower the risk of mortality in patients with moderate to severe COVID-19 [18]. However, ICU stay in tocilizumab group was found to be shorter than in the control group. In the observational studies stated above, different dosages (single or double) and subpopulations were investigated (moderate or severe or critically ill patients).

According to our knowledge, this study is the first large multicenter cohort observational study in Egypt to examine the role of tocilizumab in COVID-19 infection. We enrolled 740 COVID-19 patients, and the findings showed that there was no statistically significant difference in mortality between the tocilizumab group and the control group.

In particular, recent RCTs have not demonstrated a decrease in mortality in COVID-19 patients receiving tocilizumab [19,20,21]. Regardless of the time of administration, the number of tocilizumab doses administered, or the subset of critically ill patients admitted to the ICU, our results are consistent with those from the RCTs and demonstrate no benefit in the mortality in COVID-19 patients treated with tocilizumab. Our findings may also be consistent with earlier RCT and observational studies. A total of 51.7% of patients in the control group and 46.4% of those who received tocilizumab had been discharged from the hospital alive. Our discharge rate was similar to the study conducted by Somers et al. [22]. Indeed, we reported a high mortality rate of 53.6% compared to the control group, which is consistent with a prior study that assessed the association between tocilizumab and mortality in COVID-19 patients conducted by Campochiaro et al., who found no appreciable change in the mortality in tocilizumab-treated patients [23]. On the other hand, tocilizumab has been associated with a reduced risk of death and hospital-related mortality, according to a number of other studies [16,17,24,25,26,27,28,29].

Overall, results from different studies on the mortality rate associated with tocilizumab treatment in COVID-19 patients are contradictory, most likely as a result of disparate study designs and clinical severity classification heterogeneity. It should be noted that comparisons between observational studies are challenging due to variations in baseline disease classification heterogeneity, participant clinical severity heterogeneity, non-standardized timing, dosage, and administration of tocilizumab, low statistical power resulting from a small participant pool, lack of standardized care in the control groups, and a lack of corticosteroid effect evaluation [30]. Indeed, we contrasted the respiratory support that we offered to hospitalized patients at all participating hospitals between those who were discharged and those who sadly died. It was observed that non-rebreathing oxygen support devices, CPAP, and mechanical ventilation (MV) had the highest mortality rates compared to simple oxygen masks and nasal canula.

The incidence of serious infections is the greatest concern with tocilizumab therapy [31]. In this study, patients in the tocilizumab group required a median hospital stay of 10 days compared to 8 days in the control group. This difference can be attributed to the secondary bacterial infections, which necessitate additional antibiotic medication and decrease patient oxygenation.

The multicenter design of this trial, careful data quality monitoring, and a homogeneous target population of patients with moderate and severe pneumonia are the strengths of the study.

## 5. Conclusions

Our cohort of patients with moderate and severe COVID-19 did not assert a reduction in the risk of mortality or the length of stay (LOS). To evaluate the potential role of tocilizumab in selected patients such as ICU patients, additional RCTs are required.

## 6. Limitations

Even though we imputed missing data using modern techniques, the laboratory variables had some incomplete data. IL-6 serum concentrations were not frequently available; thus, this study did not evaluate their potential utility or how they might have predicted a patient’s response to tocilizumab.

Additionally, in the retrospective design of the research there were some possible confounders from the concomitant application of multiple interventions at the same time of TCZ administration. So, the results need confirmation by randomized controlled trials.

## Figures and Tables

**Figure 1 medicina-59-00053-f001:**
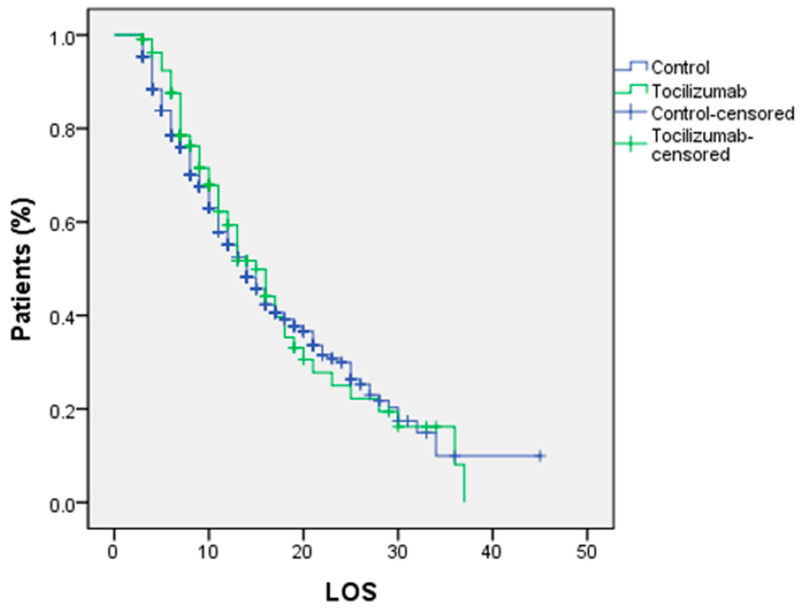
Kaplan–Meier curve of the survival rate among patients of the study groups (log rank test *p* < 0.669).

**Table 1 medicina-59-00053-t001:** Baseline clinical variables of the study population.

Clinical Variables		Tocilizumab (*n* = 110) (1)	Control (*n* = 630) (1)	*p* Value (2)
Sex	Male	68 (61.8%)	283 (44.9%)	0.01
	Female	42 (38.2%)	347 (55.1%)	
	Age	58.5 (17.5)	63(17)	0.127
COVID Severity	Moderate	69 (62.7%)	441 (70%)	0.128
	Severe	41 (37.3%)	189 (30%)	
Comorbidities	Diabetes Mellitus	60 (54.5%)	316 (50.2%)	0.396
	Cardiovascular Diseases	17 (15.5%)	171 (27.1%)	0.009
	Hypertension	54 (49.1%)	336 (53.3%)	0.411
	Chronic liver Diseases	6 (5.5%)	36 (5.7%)	0.913
	Chronic Kidney Diseases	3 (2.7%)	50 (7.9%)	0.051
	Asthma	14 (12.7%)	80 (12.7%)	0.993
	Smoking	6 (5.5%)	29 (4.6%)	0.0698

(1) Median (IQR); *n* (%); (2) Mann Whitney test; Pearson’s Chi-squared test.

**Table 2 medicina-59-00053-t002:** Baseline laboratory variables of the study population.

Laboratory Variables	Tocilizumab (*n* = 110) (1)	Control (*n* = 630) (1)	*p* Value (2)
C-reactive protein (CRP)	70 (41–74)	44.5 (40–48.9)	<0.001
Ferritin	882 (535–882)	681 (625–681)	<0.001
D Dimer	151 (77–249)	151 (95–204)	0.705
Lactate dehydrogenase (LDH)	668.3 (621–759)	707.5 (694–720)	<0.001
Aspartate transaminase (AST)	43.5 (39–44)	47.5 (33–48)	0.001
Alanine transaminase (ALT)	39.5 (30–40)	52 (28–52)	<0.001
Urea	50.5 (34.75–63)	64 (44–76)	0.001
Creatinine	1.2 (−1–2)	1.2 (1–2)	0.073
Complete blood count (CBC) Normal	23 (20.9%)	124 (19.7%)	0.766
Monocytosis	29 (26.4%)	95 (15.1%)	0.003
Lymphopenia	69 (62.7%)	334 (53%)	0.059
lymphocytosis	8 (7.3%)	32 (5.1%)	0.348
Neutrophilia	51 (46.4%)	214 (34%)	0.012
Neutropenia	8 (7.3%)	12 (1.9%)	0.005 F
Thrombocytopenia	18 (16.4%)	107(17%)	0.873
Thrombocytosis	4 (3.6%)	30 (4.8%)	0.603
Leukopenia	6 (5.5%)	24 (3.8%)	0.603 F
Leukocytosis	48 (43.6%)	229 (36.3%)	0.145

(1) Median (IQR); *n* (%); (2) Mann–Whitney test; Pearson’s Chi-squared test; F = Fisher’s exact test.

**Table 3 medicina-59-00053-t003:** Medication administered during hospitalization to the study population.

Medication	Tocilizumab (*n* = 110) (1)	Control (*n* = 630)	*p* Value (2)
Antivirals	34 (30.9%)	245 (38.9%)	0.111 C
Hydroxychloroquine	38 (34.5%)	48 (7.2%)	0.001 C
Ivermectin	77 (70%)	368 (58%)	0.022 C
Anticoagulants	110 (100%)	609 (96%)	0.058 F
Corticosteroids	110 (100%)	588 (93%).	0.002 F
Convalescent Plasma	3 (2.7%)	3 (0.5%)	0.046 F
Antibiotics	107 (97.3%)	621 (98%)	0.401 F
Antifungals	0 (0%)	35 (5.6%)	0.006 F

(1) *n* (%); (2) F = Fisher’s exact test; C = Pearson’s Chi-squared test.

**Table 4 medicina-59-00053-t004:** Clinical investigations at hospital admission of the study population.

	Clinical Investigations	Tocilizumab (*n* = 110) (1)	Control (*n* = 630) (1)	*p* Value (2)
CT Scan Findings	Ground glass opacity	53 (48.2%)	352 (55.9%)	0.001 F
	Infiltrations	2 (1%)	19 (3%)	
	Major abnormality	14 (12.7%)	80 (12.7%)	
	Mixed ground glass opacity and Infiltrations	37 (33.6%)	66 (10.5%)	
	Patchy shadowing	4 (3.6%)	113 (17.9%)	

(1) *n* (%); (2) F = Fisher’s exact test.

**Table 5 medicina-59-00053-t005:** Clinical outcomes evaluation between the two groups.

Outcome	Tocilizumab (*n* = 110) (1)	Control (*n* = 630) (1)	*p* Value (2)
Primary Outcome			
Death (Mortality)	59 (53.6%)	304 (48.3%)	0.297
Discharged alive	51 (46.4%)	326 (51.7%)	
Secondary Outcome			
Length of stay (Days) (3)	10 (7–15)	8 (5–14)	0.045

(1) *n* (%); (2) Pearson’s Chi-squared test, (3) Median (IQR); Mann–Whitney test.

**Table 6 medicina-59-00053-t006:** Distribution of respiratory support at the time of hospitalization regarding the primary outcome.

Respiratory Support	Death (*n* = 363) (1)	Discharged Alive (*n* = 377) (1)	*p* Value (2)
Ordinary simple nasal cannula	6 (1.7%)	36 (9.5%)	<0.001
Simple mask	30 (8.3%)	151 (40.1%)	
Mask with reservoir (non-rebreathing mask)	115 (31.7%)	93 (24.7%)	
High flow nasal oxygen	17 (4.7%)	44 (11.7%)	
Noninvasive ventilation, Continuous positive airway pressure (CPAP)	125 (34.4%)	47 (12.5%)	
Invasive ventilation, Mechanical ventilation (MV)	70 (19.3%)	6 (1.6%)	

(1) *n* (%); (2) Pearson’s Chi-squared test.

**Table 7 medicina-59-00053-t007:** Multivariant logistic regression analysis for factors affecting mortality.

Risk Factor	Odd Ratio	95% CI		*p*-Value
		Lower	Upper	
Sex	1.358	0.926	1.990	0.117
Smoking	0.725	0.311	1.690	0.456
Diabetes	1.065	0.726	1.561	0.749
Cardiovascular diseases	0.928	0.601	1.434	0.737
Hypertension	1.500	0.997	2.257	0.052
Chronic liver disease	1.183	0.559	2.505	0.661
Creatine kinase (CK)	0.570	0.254	1.281	0.174
Asthma	0.896	0.516	1.557	0.698
COVID Severity	1.536	1.022	2.309	0.039 *
CT scan findings	1.562	0.434	5.627	0.045 *
CBC Normal	1.625	0.985	2.682	0.058
Monocytosis	0.841	0.507	1.395	0.502
Lymphopenia	0.832	0.541	1.280	0.403
Lymphocytosis	0.502	0.213	1.181	0.114
Neutrophilia	1.120	0.721	1.739	0.614
Neutropenia	0.958	0.323	2.839	0.938
Thrombocytopenia	1.210	0.745	1.967	0.441
Thrombocytosis	2.833	1.053	7.627	0.039 *
Leukopenia	0.791	0.319	1.962	0.613
Leukocytosis	0.659	0.426	1.019	0.061
Antivirals	0.739	0.501	1.089	0.126
Hydroxychloroquine	0.940	0.514	1.719	0.841
Ivermectin	1.605	1.093	2.356	0.016 *
Anticoagulants	0.542	0.172	1.710	0.297
Corticosteroids	1.826	0.808	4.129	0.148
Convalescent plasma	14.838	1.256	17.298	0.032 *
Antibiotics	0.687	0.161	2.935	0.612
Antifungals	0.242	0.090	0.650	0.005 *
Respiratory support	0.422	0.360	0.495	0.000 *
Tocilizumab	0.764	0.446	1.308	0.326

* *p*-Value ≤ 0.05 is statistically significant.

## Data Availability

Applicable upon request.

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
