# Peer review of "Efficacy of Tocilizumab in Management of COVID-19 Patients Admitted to Intensive Care Units: A Multicenter Retrospective Cohort Study"

_medicina, 2022, doi:10.3390/medicina59010053_

Round 1

Reviewer 1 Report

Abstract. There are data that, although important for the understanding of the research work (such as inflammatory mediators), are not relevant to this section. The introduction and materials and methods occupy almost 2/3 of the abstract.

Avoid using the trade name.

Introduction. Adequate in length.

Methods. Lines 75-85. Write descriptively. Same in lines 98-100.

Lines 130-141. Remove hyphens at the beginning of each sentence.

Results. Group associated figures. Avoid excessive use of figures. They complicate reading.

Discussion. Consider this reference https://doi.org/10.54034/mic.e1251 which is quite relevant to your research topic.

Reviewer 2 Report

Major concerns.

- Is the Statistical analysis used to compare only TCZ and Control groups or include the Overalls group together?
If it includes Overalls, suggest re-analysed again with only TCZ and Control groups.
Please delineate in the statistical analysis to make the readers will not confuse.

- Results. There is no body text result after the characteristics. You cannot show only Tables and/or Figures without the body text. Please revise it.

Minor concerns.

- Suggest using "COVID-19" instead of "Covid-19" or "Covid 19" throughout the manuscript. This word is an abbreviation form and must be in all capital letters.

- Suggest rewriting the 2.1.1. too 2.1.2. to the body text, not in bullet points.

- Suggest adding the country name in 2.2.2. Study setting.

- Please use only the p-value format throughout the manuscript. Do not use both "p" and "P" (with or without italicised) together in the manuscript.

- Table 1. What is the (1) Median (IQR)? It seems that Age outcomes were mean (SD). Please correct it. Moreover, please add the (SD) in Overall age.
For Age, t-test is preferred.

- Tables 2 and 6. - Table 1. What is the (1) Median (IQR)? The data seems not to show the IQR. Please revise it.

- Tables. The data was not complete. Please revise again, especially the significant figure and percentage.
For example;
1) Ferritin (TCZ) 882.5(347.3), (Control) 681(no % showed).

2) Antiviral (TCZ) 34(30.9%), (Control) 245(38.

- Table 3. Please revise the Controls column. It is duplicated.

- Table 6. What is the central value of the outcomes? (Median with IQR or mead with SD)
Suggest revising it. A Median with IQR is preferred for this data type.

- All Figures. The figure seems wrong aspect ratio. Please use the real aspect ratio from the software.

- Figures 1 and 2. The outcomes are the same variable.
1) 2 Figures can be combined as Figure 1a and 1b.
2) Please rescale the y-axis to the same to make it comparable to the naked eye.

- Figures 6, 7, 8 and 9. Suggest using scatter/dot plots to make it more informative by individual data points.

Round 2

Reviewer 1 Report

The authors have followed the suggestions. Nothing more to add

Reviewer 2 Report

Thank you for your revision. The recent round seems easier to read than the previous one.

However, refer to your response following below.

"- All Figures. The figure seems wrong aspect ratio. Please use the real aspect ratio from the software.

Reply: All these figures have been removed according to other reviwer suggestion to decrease reader confusion."

I mean the aspect ratio is not true, it wider than a real ratio.

For example, Figure 1 (Kaplan–Meier Curve and log rank test for comparing the length of hospital stay (LOS)) is a real aspect ratio. If you compare it to other Figures in the previous that seems wider than normal.
I suggest fixing it. It is very informative material for your paper, don't remove it all.
